# CLOPS: Continual Learning of Physiological Signals

## Abstract

Deep learning algorithms are known to experience destructive interference when instances violate the assumption of being independent and identically distributed (i.i.d). This violation, however, is ubiquitous in clinical settings where data are streamed temporally and from a multitude of physiological sensors. To overcome this obstacle, we propose CLOPS, a replay-based continual learning strategy. In three continual learning scenarios based on three publically-available datasets, we show that CLOPS can outperform the state-of-the-art methods, GEM and MIR. Moreover, we propose end-to-end trainable parameters, which we term task-instance parameters, that can be used to quantify task difficulty and similarity. This quantification yields insights into both network interpretability and clinical applications, where task difficulty is poorly quantified.

## 1 Introduction

Many deep learning algorithms operate under the assumption that instances are independent and identically-distributed (i.i.d.). The violation of this assumption can be detrimental to the training behaviour and performance of an algorithm. The assumption of independence can be violated, for example, when data are streamed temporally from a sensor. Introducing multiple sensors in a changing environment can introduce covariate shift, arguably the 'Achilles heel' of machine learning model deployment (Quionero-Candela et al., 2009).

A plethora of realistic scenarios violate the i.i.d. assumption. This is particularly true in healthcare where the multitude of physiological sensors generate time-series recordings that may vary temporally (due to seasonal diseases; e.g. flu), across patients (due to different hospitals or hospital settings), and in their modality. Tackling the challenges posed by such scenarios is the focus of continual learning (CL) whereby a learner, when exposed to tasks in a sequential manner, is expected to perform well on current tasks without compromising performance on previously seen tasks. The outcome is a single algorithm that can reliably solve a multitude of tasks. However, most, if not all, research in this field has been limited to a small handful of imaging datasets (Lopez-Paz & Ranzato, 2017; Aljundi et al., 2019b;a). Although understandable from a benchmarking perspective, such research fails to explore the utility of continual learning methodologies in more realistic healthcare scenarios (Farquhar & Gal, 2018). To the best of our knowledge, we are the first to explore and propose a CL approach in the context of physiological signals. The dynamic and chaotic environment that characterizes healthcare necessitates the availability of algorithms that are dynamically reliable; those that can adapt to potential covariate shift without catastrophically forgetting how to perform tasks from the past. Such dynamic reliability implies that algorithms no longer needs to be retrained on data or tasks to which it has been exposed in the past, thus improving its data-efficiency. Secondly, algorithms that perform consistently well across a multitude of tasks are more trustworthy, a desirable trait sought by medical professionals (Spiegelhalter, 2020).

**Our Contributions.** In this paper, we propose a replay-based continual learning methodology that is based on the following:

1. **Importance-guided storage:** task-instance parameters, a scalar corresponding to each instance in each task, as informative signals for *loss-weighting* and *buffer-storage*.
2. **Uncertainty-based acquisition:** an active learning inspired methodology that determines the degree of informativeness of an instance and thus acts as a *buffer-acquisition* mechanism.

## 2 RELATED WORK

**Continual learning (CL)** approaches have resurfaced in recent years (Parisi et al., 2019). Those similar to ours comprise memory-based methods such as iCaRL (Rebuffi et al., 2017), CLEAR (Rolnick et al., 2019), GEM (Lopez-Paz & Ranzato, 2017), and aGEM (Chaudhry et al., 2018). In contrast to our work, the latter two methods naively populate their replay buffer with the last *m* examples observed for a particular task. Isele & Cosgun (2018) and Aljundi et al. (2019b) employ a more sophisticated buffer-storage strategy where a quadratic programming problem is solved in the absence of task boundaries. Aljundi et al. (2019a) introduce MIR whereby instances are stored using reservoir sampling and sampled according to whether they incur the greatest change in loss if parameters were to be updated on the subsequent task. This approach is computationally expensive, requiring multiple forward and backward passes per batch. The application of CL in the medical domain is limited to that of Lenga et al. (2020) wherein existing methodologies are implemented on chest X-ray datasets. In contrast to previous research that independently investigates buffer-storage and acquisition strategies, we focus on a *dual* storage and acquisition strategy.

**Active learning (AL) in healthcare** has observed increased interest in recent years, with a review of methodologies provided by Settles (2009). For example, Gong et al. (2019) propose a Bayesian deep latent Gaussian model to acquire important features from electronic health record (EHR) data in MIMIC (Johnson et al., 2016) to improve mortality prediction. In dealing with EHR data, Chen et al. (2013) use the distance of unlabelled samples from the hyperplane in an SVM to acquire datapoints. Wang et al. (2019) implement an RNN to acquire ECG samples during training. Zhou et al. (2017) perform transfer learning in conjunction with a convolutional neural network to acquire biomedical images in an online manner. Smailagic et al. (2018; 2019) actively acquire unannotated medical images by measuring their distance in a latent space to images in the training set. Such similarity metrics, however, are sensitive to the amount of available labelled training data. Gal et al. (2017) adopt BALD (Houlsby et al., 2011) with Monte Carlo Dropout to acquire instances that maximize the Jensen-Shannon divergence (JSD) across MC samples. To the best of our knowledge, we are the first to employ AL-inspired acquisition functions in the context of CL.

## 3 BACKGROUND

### 3.1 CONTINUAL LEARNING

In this work, we consider a learner, $f_\omega : x_\mathcal{T} \in \mathbb{R}^m \to y_\mathcal{T} \in \mathbb{R}^c$, parameterized by $\omega$, that maps an $m$-dimensional input, $x_\mathcal{T}$, to a $c$-dimensional output, $y_\mathcal{T}$, where $c$ is the number of classes, for each task $\mathcal{T} \in [1 \dots N]$. This learner is exposed to new tasks in a sequential manner once previously-tackled tasks are mastered. In this paper, we formulate our tasks based on a modification of the three-tier categorization proposed by van de Ven & Tolias (2019). In our learning scenarios (see Fig. 1), a network is sequentially tasked with solving a binary classification problem in response to data from mutually-exclusive pairs of classes **Class Incremental Learning (Class-IL)**, multi-class classification problem in response to data collected at different times of the year (e.g., winter

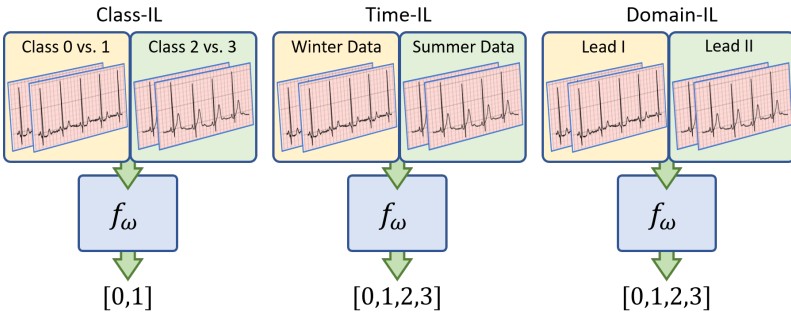

Figure 1: Illustration of the three continual learning scenarios. A network is sequentially exposed to tasks **(Class-IL)** with mutually-exclusive pairs of classes, **(Time-IL)** with data collected at different times of the year, and **(Domain-IL)** with data from different input modalities.

and summer) **Time Incremental Learning (Time-IL)**, and a multi-class classification problem in response to inputs with a different modality **Domain Incremental Learning (Domain-IL)**. In the aforementioned cases, task identities are *absent* during both training and testing and neural architectures are single-headed.

## 4 METHODS

The two ideas underlying our proposal are the storage of instances into *and* the acquisition of instances from a buffer such that destructive interference is mitigated. We describe these in more detail below.

### 4.1 IMPORTANCE-GUIDED BUFFER STORAGE

We aim to populate a buffer, $\mathcal{D}_B$, of finite size, $\mathcal{M}$, with instances from the current task that are considered important. To quantify importance, we learn parameters, entitled task-instance parameters, $\beta_{i\mathcal{T}}$, associated with each instance, $x_{i\mathcal{T}}$, in each task, $\mathcal{T}$. These parameters play a dual role.

#### 4.1.1 LOSS-WEIGHTING MECHANISM

For the current task, $k$, and its associated data, $\mathcal{D}_k$, we incorporate $\beta$ as a coefficient of the loss, $\mathcal{L}_{ik}$, incurred for each instance, $x_{ik} \in \mathcal{D}_k$. For a mini-batch of size, $B$, that consists of $B_k$ instances from the current task, the objective function is shown in Eq. 1. We can learn the values of $\beta_{ik}$ via gradient descent, with some learning rate, $\eta$, as shown in Eq. 2.

$$\mathcal{L} = \frac{1}{B_k} \sum_{i=1}^{B_k} \beta_{ik} \mathcal{L}_{ik} \qquad (1) \qquad\qquad \beta_{ik} \leftarrow \beta_{ik} - \eta \frac{\partial \mathcal{L}}{\partial \beta_{ik}} \qquad (2)$$

Note that $\frac{\partial \mathcal{L}}{\partial \beta_{ik}} = \mathcal{L}_{ik} > 0$. This suggests that instances that are hard to classify ($\uparrow \mathcal{L}_{ik}$) will exhibit $\downarrow \beta_{ik}$. From this perspective, $\beta_{ik}$ can be viewed as a proxy for instance difficulty. However, as presented, $\beta_{ik} \rightarrow 0$ as training progresses, an observation we confirmed empirically. Since $\beta_{ik}$ is the coefficient of the loss, $\mathcal{L}_{ik}$, this implies that the network will quickly be unable to learn from the data. To avoid this behaviour, we initialize $\beta_{ik} = 1$ in order to emulate a standard loss function and introduce a regularization term to penalize its undesirable and rapid decay toward zero. As a result, our modified objective function is:

$$\mathcal{L}_{\text{current}} = \frac{1}{B_k} \sum_{i=1}^{B_k} \beta_{ik} \mathcal{L}_{ik} + \lambda(\beta_{ik} - 1)^2 \qquad (3)$$

When $k > 1$, we replay instances from previous tasks by using a replay buffer (see Sec. 4.2 for replay mechanism). These replayed instances incur a loss $\mathcal{L}_{ij} \; \forall \, j \in [1 \ldots k-1]$. We decide to not weight these instances, in contrast to what we perform to instances from the current task (see Appendix K).

$$\mathcal{L}_{\text{replay}} = \frac{1}{B - B_k} \sum_{j=1}^{k-1} \sum_{i}^{B_j} \mathcal{L}_{ij} \qquad (4) \qquad\qquad \mathcal{L} = \mathcal{L}_{\text{current}} + \mathcal{L}_{\text{replay}} \qquad (5)$$

#### 4.1.2 BUFFER-STORAGE MECHANISM

We leverage $\beta$, as a proxy for instance difficulty, to store instances into the buffer. To describe the intuition behind this process, we illustrate, in Fig. 2, the trajectory of $\beta_{1k}$ and $\beta_{2k}$ associated with two instances, $x_{1k}$ and $x_{2k}$, while training on the current task, $k$, for $\tau = 20$ epochs. In selecting instances for storage into the buffer, we can 1) retrieve their corresponding $\beta$ values at the *conclusion* of the task, i.e., at $\beta(t = 20)$, 2) rank all instances based on these $\beta$ values, and 3) acquire the top $b$ fraction of instances. This approach, however, can lead to *erroneous* estimates of the relative difficulty of instances, as explained next.

In Fig. 2, we see that $\beta_{2k} > \beta_{1k}$ for the majority of the training process, indicating that $x_{2k}$ had been easier to classify than $x_{1k}$. The swap in the ranking of these $\beta$ values that occurs towards the end of training in addition to myopically looking at $\beta(t = 20)$ would *erroneously* make us believe that the

opposite was true. Such convergence or swapping of $\beta$ values has also been observed by Saxena et al. (2019). As a result, the reliability of $\beta$ as a proxy of instance difficulty is eroded.

To maintain the reliability of this proxy, we propose to *track* the $\beta$ values after each training epoch, $t$, until the final epoch, $\tau$, for the task at hand and calculate the area under these tracked values. We do so by using the trapezoidal rule as shown in Eq. 6. We explored several variants of the storage function and found the proposed form to work best (see Appendix H). At $t = \tau$, we rank the instances in descending order of $s_{ik}$ (easy to hard) as we found this preferable to the opposite order (see Appendix I), select the top $b$ fraction, and store them into the buffer, of which each task is allotted a fixed portion. The higher the value of the storage fraction, $b$, the more likely it is that the buffer will contain representative instances and thus mitigate forgetting, however this comes at an increased computational cost.

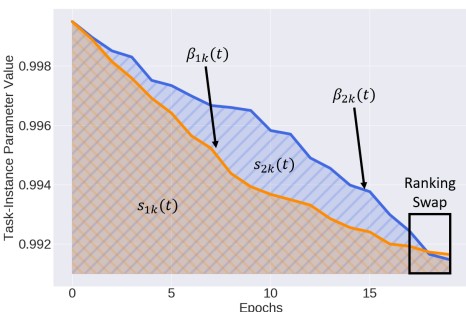

Figure 2: Trajectory of $\beta_{1k}$ and $\beta_{2k}$ on task $k$. Ranking instances based on $\beta(t = 20)$ leads to erroneous estimates of their relative difficulty. We propose to rank instances based on the area under the trajectory of $\beta$, denoted as $s_{ik}$.

$$s_{ik} = \int_0^\tau \beta_{ik}(t)dt \approx \sum_{t=0}^\tau \left( \frac{\beta_{ik}(t + \Delta t) + \beta_{ik}(t)}{2} \right) \Delta t \qquad (6)$$

## 4.2 Uncertainty-based Buffer Acquisition

The acquisition of instances that a learner is uncertain about is likely to benefit training (Zhu, 2005). This is the premise of uncertainty-based acquisition functions such as BALD (Houlsby et al., 2011; Gal & Ghahramani, 2016). We now outline how to exploit this premise for buffer acquisition.

At epoch number, $\tau_{MC}$, referred to as Monte Carlo (MC) epochs, each of the $M$ instances, $x \sim \mathcal{D}_B$, is passed through the network and exposed to a stochastic binary dropout mask to generate an output, $p(y|x, \omega) \in \mathbb{R}^C$. This is repeated $T$ times to form a matrix, $G \in \mathbb{R}^{MxTxC}$. An acquisition function, such as $\mathrm{BALD}_{MCD}$, is thus a function $\mathcal{F} : \mathbb{R}^{MxTxC} \to \mathbb{R}^M$.

$$\mathrm{BALD}_{MCD} = \mathrm{JSD}(p_1, p_2, \ldots, p_T) = \mathrm{H}\left(p(y|x)\right) - \mathbb{E}_{p(\omega|D_{train})}\left[\mathrm{H}\left(p(y|x, \hat{\omega})\right)\right] \qquad (7)$$

where $\mathrm{H}(p(y|x))$ represents the entropy of the network outputs averaged across the MC samples, and $\hat{\omega} \sim p(\omega|D_{train})$ as in Gal & Ghahramani (2016). At sample epochs, $\tau_S$, we rank instances in descending order of $\mathrm{BALD}_{MCD}$ and acquire the top $a$ fraction from each task in the buffer. A higher value of this acquisition fraction, $a$, implies more instances are acquired. Although this may not guarantee improvement in performance, it does guarantee increased training overhead. Nonetheless, the intuition is that by acquiring instances, from previous tasks, to which a network is most confused, it can be nudged to avoid destructive interference in a data-efficient manner. We outline the entire training procedure in Algorithms 1-4 in Appendix A.

# 5 Experimental Design

## 5.1 Datasets

We conduct experiments[1] in PyTorch (Paszke et al., 2019). Given our emphasis on healthcare, we evaluate our approach on three publically-available datasets that include physiological time-series data such as the electrocardiogram (ECG) alongside cardiac arrhythmia labels. We use $\mathcal{D}_1$ = **Cardiology ECG** (Hannun et al., 2019) (12-way), $\mathcal{D}_2$ = **Chapman ECG** (Zheng et al., 2020) (4-way), and $\mathcal{D}_3$ = **PhysioNet 2020 ECG** (Perez Alday et al., 2020) (9-way, multi-label). Further details regarding the datasets and network architecture can be found in Appendix C.

---

[1]Our code is available at: https://tinyurl.com/CLOPSSubmission

## 5.2 CONTINUAL LEARNING SCENARIOS

Here, we outline the three primary continual learning scenarios we use for our experiments. In **Class-IL**, $\mathcal{D}_1$ is split according to mutually-exclusive pairs of classes $[0, 1]$, $[2, 3]$, $[4, 5]$, $[6, 7]$, $[8, 9]$, and $[10, 11]$. This scenario allows us to evaluate the sensitivity of a network to new classes. In **Time-IL**, $\mathcal{D}_2$ is split into three tasks; Term 1, Term 2, and Term 3 corresponding to mutually-exclusive times of the year during which patient data were collected. This scenario allows us to evaluate the effect of temporal non-stationarity on a network's performance. Lastly, in **Domain-IL**, $\mathcal{D}_3$ is split according to the 12 leads of an ECG; 12 different projections of the same electrical signal generated by the heart. This scenario allows us to evaluate how robust a network is to the input distribution.

## 5.3 BASELINE METHODS

We compare our proposed method to the following. **Multi-Task Learning (MTL)** (Caruana, 1993) is a strategy whereby all datasets are assumed to be available at the same time and thus can be simultaneously used for training. Although this assumption may not hold in clinical settings due to the nature of data collection, privacy or memory constraints, it is nonetheless a strong baseline. **Fine-tuning** is a strategy that involves updating all parameters when training on subsequent tasks as they arrive without explicitly dealing with catastrophic forgetting. We also adapt two replay-based methods for our scenarios. **GEM** (Lopez-Paz & Ranzato, 2017) solves a quadratic programming problem to generate parameter gradients that do not increase the loss incurred by replayed instances. **MIR** (Aljundi et al., 2019a) replays instances from a buffer that incur the greatest change in loss given a parameter pseudo-update. Details on how these methods were adapted are found in Appendix C.

## 5.4 EVALUATION METRICS

To evaluate our methods, we exploit metrics suggested by Lopez-Paz & Ranzato (2017) such as average AUC and Backward Weight Transfer (BWT). We also propose two additional evaluation metrics that provide us with a more fine-grained analysis of learning strategies.

**t-Step Backward Weight Transfer.** To determine how performance changes 't-steps into the future', we propose $\mathrm{BWT}_t$ which evaluates the performance of the network on a previously-seen task, after having trained on t-tasks after it.

$$\mathrm{BWT}_t = \frac{1}{N-t} \sum_{j=1}^{N-t} \mathrm{R}_j^{j+t} - \mathrm{R}_j^j \tag{8}$$

**Lambda Backward Weight Transfer.** We extend $\mathrm{BWT}_t$ to all time-steps, $t$, to generate $\mathrm{BWT}_\lambda$. As a result, we can identify improvements in methodology at the task-level.

$$\mathrm{BWT}_\lambda = \frac{1}{N-1} \sum_{j=1}^{N-1} \left[ \frac{1}{N-j} \sum_{t=1}^{N-j} \mathrm{R}_j^{j+t} - \mathrm{R}_j^j \right] \tag{9}$$

## 5.5 HYPERPARAMETERS

Depending on the continual learning scenario, we chose $\tau = 20$ or $40$, as we found that to achieve strong performance on the respective validation sets. We chose $\tau_{MC} = 40 + n$ and the sample epochs $\tau_S = 41 + n$ where $n \in \mathbb{N}^+$ in order to sample data from the buffer at every epoch following the first task. The values must satisfy $\tau_S \geq \tau_{MC} > \tau$. For computational reasons, we chose the storage fraction $b = 0.1$ of the size of the training dataset and the acquisition fraction $a = 0.25$ of the number of samples per task in the buffer. To calculate the acquisition function, we chose the number of Monte Carlo samples, $T = 20$. We chose the regularization coefficient, $\lambda = 10$. We also explore the effect of changing these values on performance (see Appendices L and M).

# 6 EXPERIMENTAL RESULTS

## 6.1 CLASS-IL

Destructive interference is notorious amongst neural networks. In this section, we quantify such interference when learners are exposed to tasks involving novel classes. In Fig. 3a, we illustrate the AUC achieved on sequential binary classification tasks. We find that destructive interference is prevalent. For example, the network quickly forgets how to perform task $[0 - 1]$ once exposed to data from task $[2 - 3]$. This can be seen by the AUC $\approx 0.92 \rightarrow 0.30$. The final performance of the network for that particular task (AUC $\approx 0.78$) is also lower than that maximally-achieved. In Fig. 3b, we show that CLOPS alleviates this interference. This can be seen by the absence of significant drops in AUC and higher final performance for all tasks relative to the fine-tuning strategy.

In Table 1, we compare the performance of the CL strategies in the Class-IL scenario. We find that CLOPS outperforms MTL (AUC = 0.796 vs. 0.701), which is a seemingly non-intuitive finding. We hypothesize that this finding is due to positive weight transfer brought about by a curriculum wherein sequential tasks of different levels of difficulty can improve generalization performance (Bengio et al., 2009). We explore this hypothesis further in Sec. 6.5. We also find that CLOPS outperforms state-of-the-art methods, GEM and MIR, in terms of generalization performance and exhibits constructive interference. For example, CLOPS and MIR achieve an AUC = 0.796 and 0.753, respectively. Moreover, BWT = 0.053 and 0.009 for these two methods, respectively. Such a finding underscores the ability of CLOPS to deal with tasks involving novel classes. We also show that CLOPS is robust to task order (see Appendix F).

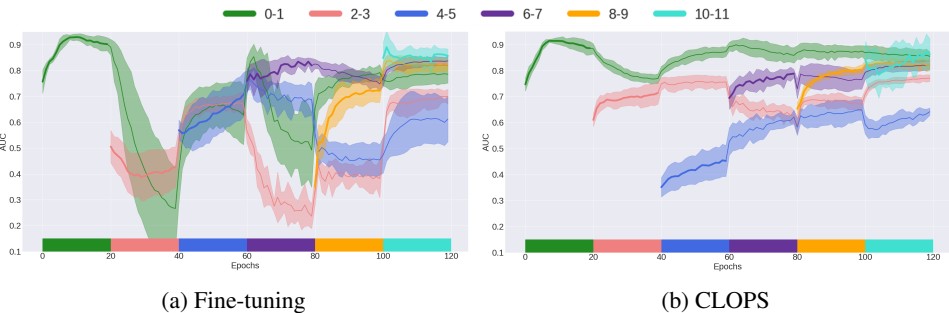

|  (a) Fine-tuning  |  (b) CLOPS  |

Figure 3: Mean validation AUC of the a) fine-tuning and b) CLOPS strategy ($b = 0.25$ and $a = 0.50$) in the Class-IL scenario. Coloured blocks indicate tasks on which the learner is currently being trained. The shaded area represents one standard deviation across five seeds.

Table 1: Performance of CL strategies in the Class-IL scenario. Storage and acquisition fractions are $b = 0.25$ and $a = 0.50$, respectively. Mean and standard deviation are shown across five seeds.

| Method | Average AUC | BWT | BWT$_t$ | BWT$_\lambda$ |
|---|---|---|---|---|
| MTL | $0.701 \pm 0.014$ | - | - | - |
| Fine-tuning | $0.770 \pm 0.020$ | $0.037 \pm 0.037$ | $(0.076) \pm 0.064$ | $(0.176) \pm 0.080$ |
| *Replay-based Methods* | | | | |
| GEM | $0.544 \pm 0.031$ | $(0.024) \pm 0.028$ | $(0.046) \pm 0.017$ | $(0.175) \pm 0.021$ |
| MIR | $0.753 \pm 0.014$ | $0.009 \pm 0.018$ | $0.001 \pm 0.025$ | $(0.046) \pm 0.022$ |
| CLOPS | $\mathbf{0.796 \pm 0.013}$ | $\mathbf{0.053 \pm 0.023}$ | $\mathbf{0.018 \pm 0.010}$ | $\mathbf{0.008 \pm 0.016}$ |

## 6.2 TIME-IL

Environmental changes within healthcare can introduce seasonal shift into datasets. In this section, we quantify the effect of such a shift on learners. In Fig. 4a, we illustrate the AUC achieved on tasks with seasonally-shifted data.

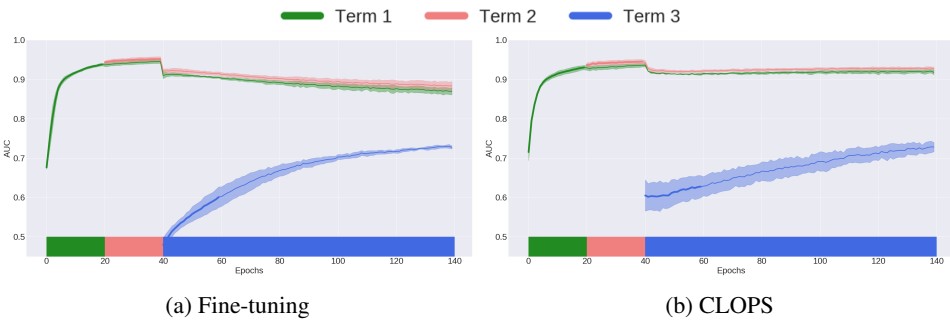

(a) Fine-tuning               (b) CLOPS

Figure 4: Mean validation AUC of the (a) fine-tuning and (b) CLOPS strategy in the Time-IL scenario. Coloured blocks indicate tasks on which the learner is currently being trained. The shaded area represents one standard deviation from the mean across five seeds.

In this scenario, we find that CLOPS is capable of achieving forward weight transfer (FWT). For example, in Figs. 4a and 4b, CLOPS achieves an AUC $\approx 0.62$ after one epoch of training on task Term 3, a value that the fine-tuning strategy only achieves after 20 epochs, signalling a 20-fold reduction in training time. We attribute this FWT to the loss-weighting role played by the task-instance parameters. By placing greater emphasis on more useful instances, the generalization performance of the network is improved. We also find that CLOPS exhibits reduced catastrophic forgetting relative to fine-tuning. For example, performance on tasks Term 1 and Term 2 is maintained at AUC $> 0.90$ when training on task Term 3. We do not observe this for the fine-tuning setup.

## 6.3 DOMAIN-IL

So far, we have shown the potential of CLOPS to alleviate destructive interference and allow for forward weight transfer. In this section, and in Table 2, we illustrate the performance of the CL strategies in the Domain-IL scenario. We show that CLOPS outperforms state-of-the-art methods. For example, CLOPS and MIR achieve an AUC $= 0.731$ and $0.716$, respectively. CLOPS is also better at mitigating destructive interference, as shown by BWT $= (0.011)$ and $(0.022)$, respectively. We provide an explanation for such performance by conducting ablation studies in the next section.

Table 2: Performance of CL strategies in the Domain-IL scenario. Storage and acquisition fractions are $b = 0.25$ and $a = 0.50$, respectively. Mean and standard deviation are shown across five seeds.

| Method | Average AUC | BWT | BWT$_t$ | BWT$_\lambda$ |
|---|---|---|---|---|
| MTL | $0.730 \pm 0.016$ | - | - | - |
| Fine-tuning | $0.687 \pm 0.007$ | $(0.041) \pm 0.008$ | $(0.047) \pm 0.004$ | $(0.070) \pm 0.007$ |
| *Replay-based Methods* | | | | |
| GEM | $0.502 \pm 0.012$ | $(0.025) \pm 0.008$ | $\mathbf{0.004 \pm 0.010}$ | $(0.046) \pm 0.021$ |
| MIR | $0.716 \pm 0.011$ | $(0.022) \pm 0.011$ | $(0.013) \pm 0.004$ | $(0.019) \pm 0.006$ |
| CLOPS | $\mathbf{0.731 \pm 0.001}$ | $\mathbf{(0.011) \pm 0.002}$ | $(0.020) \pm 0.004$ | $(0.019) \pm 0.009$ |

## 6.4 EFFECT OF TASK-INSTANCE PARAMETERS, $\beta$, AND ACQUISITION FUNCTION, $\alpha$

To better understand the root cause of CLOPS' benefits, we conduct three ablation studies: 1) **Random Storage** dispenses with task-instance parameters and instead randomly stores instances into the buffer, 2) **Random Acquisition** dispenses with acquisition functions and instead randomly acquires instances from the buffer, and 3) **Random Storage and Acquisition** which stores instances into, and acquires instances from, the buffer randomly. In Fig. 5, we illustrate the effect of these strategies on performance as we vary the storage fraction, $b$, and acquisition fraction, $a$.

We find that $\beta$, as a loss-weighting mechanism, benefits generalization performance. For example, in Fig. 5 (red rectangle), at $b = 1, a = 0.5$, we show that simply including the loss-weighting

mechanism $\uparrow$ AUC $\approx 12\%$. We hypothesize that this mechanism is analogous to attention being placed on instance losses and thus allows the network to learn *which* instances to exploit further. We also find that uncertainty-based acquisition functions offer significant improvements. In Fig. 5 (black rectangles), at $a = 0.1, b = 0.5$, we show that such acquisition $\uparrow$ AUC $\approx 8\%$. We arrive at the same conclusion when evaluating backward weight transfer (see Appendix J).

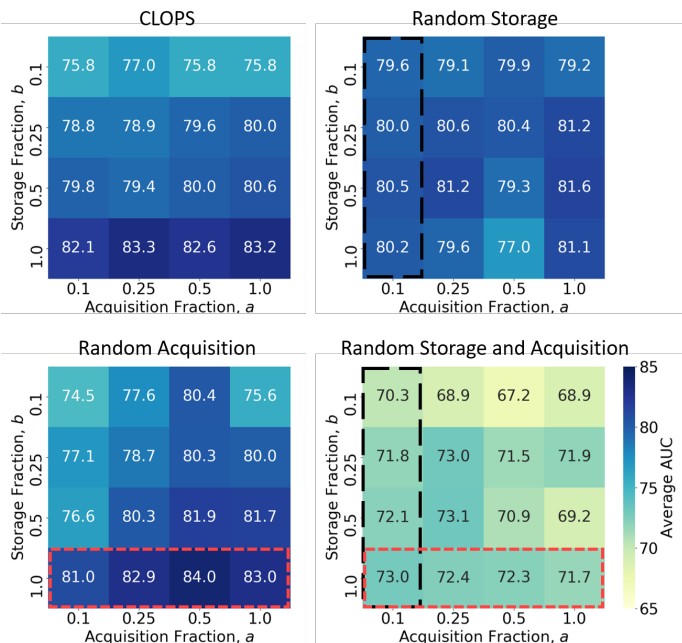

Figure 5: Mean validation AUC of four different learning strategies in the Class-IL scenario. 1) CLOPS, 2) Random Storage, 3) Random Acquisition, and 4) Random Storage and Acquisition. Results are shown as a function of storage fractions, *b*, and acquisition fractions, *a* and are an average across five seeds. We highlight the utility of (red rectangle) task-instance parameters as a loss-weighting mechanism and (black rectangle) uncertainty-based acquisition functions for the acquisition of instances from the buffer.

## 6.5 VALIDATION OF INTERPRETATION OF TASK-INSTANCE PARAMETERS, $\beta$

We claimed that instances with lower values of $\beta$, and by extension, $s$, are relatively more difficult to classify. In this section, we aim to validate this intuition. In Fig. 6, we illustrate the distribution of $s$ values corresponding to each task.

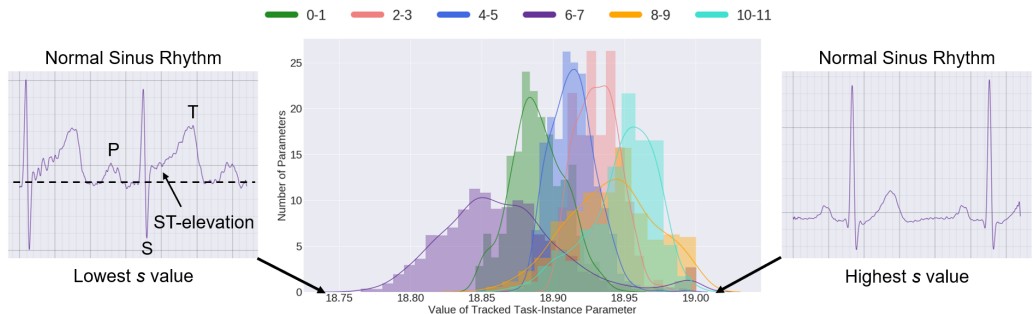

Figure 6: Distribution of the $s$ values corresponding to CLOPS ($b = 0.25$ and $a = 0.50$) in the Class-IL scenario. Each colour corresponds to a different task. The ECG recording with the lowest $s$ value is labelled as normal despite the presence of ST-elevation, a feature common in heart attacks.

We find that tasks differ in their difficulty level. For example, task $[6-7]$ is considered more difficult to solve than task $[8-9]$ as evidenced by the lower distribution mean of the former relative to the latter ($s \approx 18.85$ vs. $18.95$). After extracting the two ECG recordings associated with the lowest and highest $s$ values, we find that both belong to the same class, normal sinus rhythm. Upon closer inspection, the recording with the lower $s$ value exhibits a feature known as ST-elevation. This feature, which is characterized by the elevation of the segment between the S and T waves (deflections) of the ECG signal relative to the baseline, is typically associated with heart attacks. Mapping an ECG recording with such an abnormal feature to the normal class would have been a source of confusion for the network. We provide additional qualitative evidence in Appendix G.

We also leverage $s$ to learn a curriculum (Bengio et al., 2009). First, we fit a Gaussian, $\mathcal{N}(\mu_\mathcal{T}, \sigma_\mathcal{T}^2)$, to each of the distributions in Fig. 6. Using this information, we define the difficulty of task $\mathcal{T}$ as $d_\mathcal{T} = \frac{1}{\mu_\mathcal{T}}$ and the similarity, $S(j, k)$, between task $j$ and $k$ as shown in Eq. 10. In Fig. 7, we illustrate the resulting pairwise task similarity matrix.

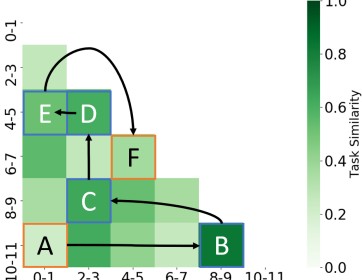

$$S(j,k) = 1 - \underbrace{\sqrt{1 - \sqrt{\frac{2\sigma_0\sigma_1}{\sigma_0^2\sigma_1^2}}e^{-\frac{1}{4}\frac{(\mu_0-\mu_1)^2}{\sigma_0^2\sigma_1^2}}}}_{\mathcal{D}_H = \text{Hellinger Distance}} \quad (10)$$

Figure 7: Similarity of tasks in the Class-IL scenario. We create a curriculum by following chaining tasks that are similar to one another.

We design a curriculum by first selecting the easiest task ($\downarrow d_\mathcal{T}$) and then creating a chain of tasks that are similar to one another as shown in Fig. 7. For an anti-curriculum, we start with the hardest task ($\uparrow d_\mathcal{T}$). In Table 3, we illustrate the performance of various curricula and find that a curriculum exhibits higher constructive interference than a random one (BWT $= 0.087$ vs. $0.053$). Such an outcome aligns well with the expectations of curriculum learning, thus helping to further validate the intuition underlying $\beta$.

Table 3: Performance of CLOPS in the Class-IL scenario with different curricula. Storage and acquisition fractions are $b = 0.25$ and $a = 0.50$, respectively. Results are shown across five seeds.

| Task Order | Average AUC | BWT | $BWT_t$ | $BWT_\lambda$ |
|---|---|---|---|---|
| Random | $\mathbf{0.796 \pm 0.013}$ | $0.053 \pm 0.023$ | $0.018 \pm 0.010$ | $0.008 \pm 0.016$ |
| Curriculum | $0.744 \pm 0.009$ | $\mathbf{0.087 \pm 0.011}$ | $\mathbf{0.038 \pm 0.021}$ | $\mathbf{0.076 \pm 0.037}$ |
| Anti-curriculum | $0.783 \pm 0.022$ | $0.058 \pm 0.016$ | $(0.013) \pm 0.013$ | $(0.003) \pm 0.014$ |

## 7 DISCUSSION AND FUTURE WORK

In this paper, we introduce a replay-based method applied to physiological signals, entitled CLOPS, to mitigate destructive interference during continual learning. CLOPS consists of an importance-guided buffer-storage and active-learning inspired buffer-acquisition mechanism. We show that CLOPS outperforms the state-of-the-art methods, GEM and MIR, on both backward and forward weight transfer. Furthermore, we propose learnable parameters, as a proxy for the difficulty with which instances are classified, which can assist with quantifying task difficulty and improving network interpretability. We now elucidate future avenues worth exploring.

**Extensions to Task Similarity.** The notion of task similarity was explored by Thrun & O'Sullivan (1996); Silver & Mercer (1996). In this work, we proposed a definition of task similarity and used it to order the presentation of tasks. The exploration of more robust definitions, their validation through domain knowledge, and their exploitation for generalization is an exciting extension.

**Predicting Destructive Interference.** Destructive interference is often dealt with in a reactive manner. By *predicting* the degree of forgetting that a network may experience once trained sequentially can help alleviate this problem in a proactive manner.

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
