# OpenReview forum: "CLOPS: Continual Learning of Physiological Signals"
_ICLR.cc/2021/Conference — Reject_

### Official Review · AnonReviewer4 · 2020-10-27
**Clearly described approach; not sure about evaluation quality**

**Rating:** 7
**Confidence:** 2

**Review:**

**Update**
I think the manuscript has been further improved now and I improve my score to 7. Also since I think results on public datasets in new domains other than images may be helpful for the wider research community. Keep in mind, as written below, that other reviewers more familiar with this research field may be more  able than me to judge the evaluation quality and whether the evaluation contains too substantial flaws to allow publication.

**Disclaimer**
Upon reviewing the manuscript, I found I am unfortunately not well-acquainted enough with the research fields (active learning/continual learning) to be able to evaluate very well what this manuscript contributes to these fields. I hope my review can still be useful.

**Summary**
The manuscript describes and evaluates a continual learning algorithm that learns per-instance scalars beta that weight the loss of each instance. These betas are optimized together with all the other parameters, using an additional regularization term that prevents all betas to be optimized to zero. The betas averaged over time are used to decide which instances get into the replay buffer (easier ones are used). Additionally, to decide which instances  from the replay buffer to train on, Monte Carlo Dropout is used to estimate the uncertainties of the network on each instance and more uncertain instances are then more likely to be sampled for training. The approach is evaluated on three healthcare datasets, in either a setting where sequentially different tasks are learned by the network, the same task is learned with data sequentially used across time (affected by seasonal changes), or the same task is learned while sequentially shifting the domain/input distribution in a controlled way (different projections of same ECG signal). The proposed approach outperforms two baselines with regard to performance on previously-trained-on tasks. Ablation studies show the role of the storage strategy and the acquisition strategy while validating the interpretation of the betas as instance difficulties.

**Main Impression**
The approach is clearly described, it was fairly easy to understand for me, the writing is nice. The evaluation seems interesting; it would be more convincing to me if it was also evaluated on some dataset where published results of other works are available. Ablation studies are nice, the results of them could be discussed more.

**Major Points**

As written above, I would really like to see an evaluation of this approach where one can directly compare to published results by others. Reimplemented baselines always have a risk of being not tuned well enough etc. And as far as I could understand, there does not seem to be anything so healthcare-specific in the algorithm itself that comparison in other domains would not make sense, in case that is the problem.

I like the ablation studies and would have enjoyed more discussion of their results.
- It seems for b=0.1 and b=0.25, random storage always outperforms CLOPS? Why? b=0.5 also mostly outperforms.
- Similar, for a=0.5 and a=1.0, it seems that random acquisition mostly outperforms CLOPS, for a=0.25 it is still close, why?
- What happens for random acquisition and random storage? Was this tested?


 **Minor Points**

*Some parts of introduction I didn’t agree or find too broad*

“Deep learning algorithms operate under the assumption that instances are independent and identically-distributed (i.i.d.). “
-> not really specific to deep learning at all to me. Obviously one can apply deep neural networks also under different assumptions. Would either write: “Many machine learning and deep learning algorithms ...”, Or simply “Many deep learning algorithms…”

“Given the potential impact of designing such an algorithm and the machine learning community’s efforts towards achieving artificial general intelligence, research on continual learning has increased (Parisi et al., 2019).”
-> a bit too broad stuff to me, could be just removed (AGI etc.)

*Confusing beta zero part*

“In this setup, we discovered β_iT quickly decay to zero, and as a coefficient of the loss, prevents the network from learning the task”
This made me a bit confused: As per formula, if you set all betas to 0, the loss would become zero no? So it should be a direct consequence of the formula that yes, gradient descent or any optimization should set all betas to zero without any regularization/constraints? The “discovered” confused me here. Or am I misunderstanding something here?

*Figure improvements*

Figure 2: thicker curves? Hard to see

Figure 5 in appendix: legend would be helpful



As written above, I don’t feel well-acquainted enough with the literature on continual/active learning to judge the contribution of this manuscript to these fields with confidence; hope other reviewers can do this :)

---

> ### Author Response · Authors · 2020-11-18
> **Response to Reviewer 4 - Round 1**
>
> We thank the reviewer for taking the time and effort to review the manuscript and for providing us with valuable feedback. We address your comments below.
>
> **EXPLANATION OF ABLATION RESULTS**
> At b = 0.1, we find that the Random Storage approach outperforms CLOPS. This implies that a strategy, with all else being equal, that randomly chooses 10% of the instances to store into the buffer is better than one that leverages the storage function (Eq. 6) to store instances. We offer a potential explanation for this observation. Recall that our storage mechanism stores the relatively ‘easiest-to-classify’ instances in the buffer. If these ‘easiest-to-classify’ instances happen to belong predominantly to only one of the two classes from the previous task, then the buffer ends up containing data that is heavily biased towards a small subset of classes, and thus exhibits poor coverage of the data distribution from previous tasks. Therefore, downstream acquisition of these instances from the buffer might hinder performance.
>
> To explain the relative improvement in performance of Random Acquisition over CLOPS at a=0.5, we offer a similar argument based on data coverage. Recall that our acquisition strategy is based on how uncertain a network is about a particular instance. Therefore, if the network happens to be most uncertain about instances from a subset of classes, then the instances that are acquired from the buffer will be a poor reflection of the data distribution of the previous tasks. This, in turn, provides an imbalanced supervisory signal upon training and might hinder performance.
>
> **NEW RANDOM STORAGE AND ACQUISITION EXPERIMENT**
> In the modified version of the manuscript, we illustrate the results for the random storage and acquisition experiments which we just recently conducted (Sec. 6.4). We provide an explanation for our results that is focused on the utility of task-instance parameters as loss-weighting mechanisms and the utility of leveraging an acquisition function to acquire instances from the buffer.
>
> **BROAD SENTENCES**
> We have modified sentences that consist of broad generalizations.
>
> **EXPLANATION OF LOSS-WEIGHTING MECHANISM**
> We modify Sec. 4.1.1 to improve clarity. In the process, we have attempted to better explain the role of task-instance parameters as a loss-weighting mechanism and the need for a regularization term to avoid a vanishing loss term. Please inform us if this section remains unclear.
>
> **FIGURE IMPROVEMENTS**
> We have added coloured-blocks to each of the figures containing validation AUC curves to illustrate when a task is currently being trained on. This should improve the clarity of the figures. We have also annotated what used to be Fig. 5 (and is now Fig. 2) to improve clarity and help explain the intuition behind our storage function design.
>
> We hope the above responses and the modified version of the manuscript have addressed your concerns.

---

> > ### Comment · AnonReviewer4 · 2020-11-20
> > **Many issues resolved, remains the empirical comparison to published results issue**
> >
> > EXPLANATION OF ABLATION RESULTS and NEW RANDOM STORAGE AND ACQUISITION EXPERIMENT
> > Thank you for the additional discussion and experiment, I think it adds to the value of the manuscript.
> >
> > BROAD SENTENCES
> > Thank you, reads better to me now
> >
> > EXPLANATION OF LOSS-WEIGHTING MECHANISM
> > Better than before, I would maybe even prefer writing in words like “beta will go to zero” and no need to add the confirmed empirically, but these are minor details.
> >
> > FIGURE IMPROVEMENTS
> > Thanks, to me it is much more easily understandable now. You may consider whether the figures, esp. Figure 3, are understandable for persons with colorblindness. I am not colorblind myself so cannot judge, just noticed there seems to be green and something reddish used in Fig.3?
> >
> > Overall, the main unaddressed issue remaining is evaluation on published data, but I will write in the thread with Reviewer 5 to not split up discussion further.

---

### Official Review · AnonReviewer2 · 2020-10-30
**a unique combination of applying Active Learning to physiological signals for CL**

**Rating:** 7
**Confidence:** 4

**Review:**

The authors propose a learning methodology designed to offset detriments to algorithm performance that arise when instances are not i.i.d (independent and identically distributed), focusing on cases in continual learning (CL) given physiological signals. They designed a replay-based learning method that handles an instance buffer using Importance-guided Storage and Uncertainty-based Acquisition strategies. They apply their method on Class, Time and Domain types of CL, and they introduce t-Step Backward Weight Transfer and  Lambda Backward Weight Transfer methods by which to evaluate their method. They conclude with two ablation studies to explore an explanation for their method’s performance and attempt to validate their hypotheses based on these studies.

Suggestions:
* Make more explicit, in 4.1.2 (regarding equation 5), which type of task is being referred to, when comparing between prior and present tasks.
* If time permits, a figure for Experimental Results section 6.3 (to correspond to figures provided in 6.1 and 6.2) would be nice.
* A brief explanation should be included for the -1 in the regularization term in equation (2)
* In section 4.1.2, the need for a storage function was well motivated; however, the particular solution that the authors chose needs more motivation (and comparison with possible alternatives such as \beta^2. Not suggesting that this be empirically verified.)
* In Figure 2, SDs were calculated over 5 seeds. Was this low number due to the computational cost of experimentation?
* For a lay ML practitioner to appreciate the assertion made in section 6.5, "ST-elevation" needs to be defined more concretely.

---

> ### Author Response · Authors · 2020-11-18
> **Response to Reviewer 2 - Round 1**
>
> We thank the reviewer for taking the time and effort to review the manuscript and for providing us with valuable feedback. We address your comments below.
>
> **CLARIFYING EQUATION 5**
> We modify Sec. 4.1.1 to improve clarity and avoid any potential confusion that may arise based on previous and current task terminology.
>
> **FIGURE FOR SECTION 6.3**
> Although we do have this figure at hand, we had opted to relegate it to the Appendix (Appendix E.2). Its results are consistent with those shown elsewhere in the main manuscript and would have served to reaffirm our findings. In the interest of including more description of the ablation studies and validation of the task-instance parameters, we opted to keep this figure in Appendix E.2.
>
> **EXPLANATION OF REGULARIZATION TERM**
> As mentioned above, we modified Sec. 4.1.1 in order to improve clarity and address potential confusions that had arisen in that section. We include an additional equation (Eq. 2) to help illustrate the issue we are trying to circumvent by introducing the regularization term. In a nutshell, Eq. 2 suggests that instances with a larger loss value (hard-to-classify) will exhibit lower values of Bit. As presented, however, the derivative Bit = Lit > 0 implies that Bit -> 0 as training progresses, which we confirmed empirically. This implies that L -> 0 and the network is unable to learn from the data. To avoid this behaviour, we initialize Bit as 1 (as with a normal loss function) and introduce a regularization term to penalize extreme deviations from 1. Please refer to Sec. 4.1.1 for a more complete picture.
>
> **MOTIVATION FOR STORAGE FUNCTION**
> We provide additional motivation for the choice of our storage function in Sec. 4.1.2. This explanation comes alongside Fig. 2 (which was in the Appendix) and is now annotated to better complement the explanation. We also implement a variant of our storage function (Eq. 14, Appendix H), as recommended by the reviewer, and illustrate its inferior performance relative to the one we had chosen to use in the main manuscript. These results can be found in Appendix H.
>
> **NUMBER OF SEEDS**
> Each of our experiments are conducted over 5 seeds in attempt to avoid any one-off positive or negative outlier experiments and obtain a reasonable estimate of the performance of the methods. This number of seeds is quite typical in the machine learning domain and is partially motivated by the computational cost of experimentation
>
> **MEDICAL JARGON**
> We explain the importance of the ST-elevation illustrated in Fig. 6 and annotate that figure to help machine learning practitioners better understand the implications of it in this context.
>
> We hope the above responses and the modified version of the manuscript have addressed your concerns.

---

### Official Review · AnonReviewer5 · 2020-11-05
**No theoretical basis, slack empirical validation**

**Rating:** 3
**Confidence:** 4

**Review:**

## Summary
This paper proposes a replay-based continual learning method and applies it to physiological data.
The proposed method consists of two heuristics to manage the replay memory:
- Learn a weight parameter (task-instance parameter) for each data point by minimizing a special loss function, and store the data with large weight parameters.
- Periodically compute epistemic uncertainty for all data in the replay memory, and replay the data with large uncertainty.

---

## Pros

This paper points out the importance of continual learning in the medical field.

---

## Cons

### Weak contribution

Since this paper deals with the continual learning of a specific type of data, I think there are two directions in which this study can be meaningful:
- Test various CL algorithms in the new domain, determine which method performs better than others, and analyze the reason.
- Use domain knowledge to design a new algorithm that performs particularly better on the domain.

However, I think this paper belongs to neither of the two. There is no domain-specific component in the algorithm, and only two baselines are compared.

If the method is general enough to be used in other domains, I think the authors should have also tested standard CL scenarios such as Split-CIFAR10 or Split-CIFAR100.

### Lack of theoretical justification

The proposed method is mostly heuristic and does not have a theoretical basis. Especially eq. (2) seems too arbitrary. Since this paper's empirical evidence is weak, I think a theoretical analysis is necessary to support the algorithm's efficacy.

### Vague and improper definition of time incremental learning

The authors seem to propose time incremental learning for the first time, but the definition is too vague:
> Time Incremental Learning (Task-IL) - the same dataset and prediction problem are used for each task, however the time of year at which the data were collected differs from one task to another. Such seasonality is most common in healthcare applications.

The authors should specify what changes over time. However, I think there is even a bigger problem:
- If the input distribution changes, it is not different from domain incremental learning.
- If the output distribution changes for the same input, it is not continual learning. The model *should* forget old tasks and adapt to new tasks.
- If both the input and output distributions change, it is class incremental learning.

Therefore, I think time incremental learning is not a novel branch of continual learning. The corresponding experiments should better be reformulated as domain incremental learning or class incremental learning.

---

## Overall evaluation

I do not think this paper proposes a novel idea with either a solid theoretical basis or strong empirical results. Therefore, I recommend rejection.

---

## Post rebuttal
During the rebuttal, the authors failed to handle the issues that I raised. Especially, the authors did not respond to my criticisms about the strange experiments. Therefore, I stick to my initial rating.

---

> ### Author Response · Authors · 2020-11-18
> **Response to Reviewer 5 - Round 1**
>
> We thank the reviewer for taking the time and effort to review the manuscript and for providing us with valuable feedback. We address your comments below.
>
> **CONTRIBUTIONS**
> Existing continual learning algorithms are predominantly tested on the two datasets you had suggested we experiment with: Split-CIFAR10 or Split-CIFAR100. In addition, the tasks they concoct are quite unrealistic and several researchers in this domain have begun asking for realistic scenarios in which continual learning algorithms can be evaluated. You can refer to https://www.youtube.com/watch?v=N7XJ-QTEoHI at 28:00 for more details. We believe the continual learning setting is naturally found in the medical domain and such an observation motivates our entire paper. In addition to the potential clinical impact of our proposed approach, we believe it exhibits novelty in two distinct ways. First, we introduce an importance-guided buffer storage mechanism and an uncertainty-based acquisition mechanism. To the best of our knowledge, this has not been implemented before in the literature. Beyond novelty from the perspective of methodology, we also implement various state-of-the-art replayed based CL algorithms and illustrate that CLOPS outperforms those methods in three realistic CL scenarios.
>
> **EMPIRICAL EVIDENCE**
> We would like to respectfully disagree that the empirical evidence is weak. We are happy to hear your thoughts on which components of the empirical results are considered weak from your perspective. We have illustrated that CLOPS (our method) outperforms state-of-the-art replay-based CL methods (GEM and MIR) in three realistic continual learning scenarios involving three large-scale ECG datasets. Moreover, not only do we obtain strong generalization performance (i.e., Average AUC), we also manage to mitigate catastrophic forgetting (i.e., higher backward weight transfer) relative to these state-of-the-art methods.
>
> **DEFINITION OF CONTINUAL LEARNING SCENARIOS**
> We would also like to respectfully disagree that our definition of Time-IL is improper. To clarify any misunderstandings about the CL scenarios, we introduce Fig. 1 in the modified version of the manuscript. In a nutshell, and in our context Time-IL reflects a realistic scenario that is experienced in the medical domain and is defined by a multi-class classification problem in response to data collected at different times of the year (e.g., winter, summer). To use the scenario breakdown you have provided, Time-IL consists of different inputs and same outputs. If you feel this is better suited to Domain-IL, then we can introduce Time-IL as a sub-branch of Domain-IL. It just depends on how you introduce the domain shift.
>
> We refer the reviewer to the modified version of the manuscript in order to get a better understanding and appreciation of the contributions of our proposed methodology and its potential impact on the clinical workflow.
>
> We hope the above responses and the modified version of the manuscript have addressed your concerns.

---

> > ### Comment · AnonReviewer5 · 2020-11-19
> > **Re: Response to Reviewer 5 - Round 1**
> >
> > **Contributions**
> >
> > I could not agree more that applying continual learning to realistic scenarios is important. However, it is hard to agree that the proposed CL problems are *realistic*. Although the type of data is different, the scale of the scenario seems to be not that different from the existing ones (correct me if I am wrong). The youtube video mentioned by the authors points out that many CL problems do not match real-world application needs. Then, do the proposed problems match real-world application needs? Considering the size of the problems, will the practitioners actually use a CL method, rather than full re-training?
> >
> > Since I think designing a truly realistic CL scenario itself is a significant challenge, I do not want to problematize how realistic the scenario is. Instead, I asked for novel insights from the combination of CL and a new type of domain. That is why I thought the paper should be fitted into one of the two directions that I mentioned in the initial review. Although the proposed method is something new, I could not find any connection to the problem domain.
> >
> > ---
> >
> > **Empirical Evidence**
> > First, I apologize for not explaining why I thought the empirical validation is weak. I think I mistakenly deleted a whole paragraph about that while changing the ordering of some sections.
> >
> > The main reason I thought the experiments are possibly problematic is that the relative performances of baselines are strange. In every CL paper that I know, fine-tuning is always the worst and MTL is always the best. However, In table 1, fine-tuning is better than GEM, MIR, and even MTL. In table 2, fine-tuning is better than GEM and CLOPS is better than MTL. I think this strongly suggests that the baselines are poorly tuned.
> >
> > In the response to Reviewer 1, the authors claim that this is the effect of curriculum learning. In curriculum learning, however, the data distribution of the final training step is the same as the whole distribution. This is why catastrophic forgetting does not happen in curriculum learning. Then, there are two ways to interpret the strange result:
> > - The baselines are poorly tuned.
> > - The training scenario is far from continual learning.
> >
> > Either way, I think it is a critical problem.
> >
> > ---
> >
> > **Definition of Continual Learning Scenarios**
> >
> > I think Time-IL should be introduced as a sub-branch of Domain-IL.

---

> > > ### Author Response · Authors · 2020-11-19
> > > **Response to Reviewer 5 - Round 2**
> > >
> > > **CONTRIBUTIONS**
> > > Our proposed problems had been motivated by real-world clinical workflows and designed based on discussions with medical practitioners in hospital settings. As an exemplar, we provide you with a concrete example of such a clinical workflow. Let us assume we have a patient connected to multiple physiological recording devices (e.g., an ECG machine). Note that this patient can either be in a hospital bed or ambulatory (i.e., moving around while wearing a smartwatch with physiological recording capabilities). Data from such devices are streaming in an online manner as they measure the physiological activity of the patient. In the case of cardiac arrhythmia (abnormalities of the heart), this patient may experience various forms (classes) of such abnormalities while wearing the smartwatch for a certain duration of time. In this context, we need a system that is capable of identifying potentially new classes of arrhythmia that are introduced as time progresses (Class-IL), or to adapt to shifted input distributions as a result of the patient hiking near Mt. Fuji (Domain-IL). A system capable of dynamically adapting to such situations is critical as it ensures we have robust cardiac arrhythmia classification algorithms. This brings us to the connection to the problem domain that you had mentioned. Beyond the proposed method, we believe the connection of our paper to the problem domain lies in our formulation of the three continual learning scenarios themselves (Class-IL, Domain-IL, Domain-IL with temporal component).
> > >
> > > As for the size of the problems, at the time of writing, we had used the two largest publicly-available 12-lead ECG datasets. Of course, validation on much larger privately-held datasets tucked away at clinical institutions would add significant value, but as you can imagine, accessing such data is nontrivial. Having said that, we believe that the size of the dataset is not the only factor one should consider when it comes to deciding between a CL method and full re-training. Other factors include the relatively higher computational cost of full re-training, both in terms of physical resources but also the time taken to do so. A healthcare-specific factor is that of patient privacy. For example, certain regulations stipulate that some data can only be stored on servers at healthcare institutions for a certain number of months before it must be discarded. As a result, full re-training on such data is not possible. Nonetheless, we believe our formulation opens the door for healthcare institutions with access to large-scale datasets to exploit CL.
> > >
> > > **EMPIRICAL EVIDENCE**
> > > Typically, MTL does perform best and fine-tuning does perform worst. Indeed we confirm these intuitions in our experiments (apart from the Class-IL scenario). In the Class-IL scenario, we would like to clarify the differences between MTL and the remaining methods (e.g., fine-tuning, CLOPS, etc.) in attempt to further explain potential differences between the results of these methods. The cardiology dataset, which is used for the Class-IL scenario, consists of 12 cardiac arrhythmia classes. As a result, in the MTL setting, this problem is a 12-way multi-class classification problem, which is arguably quite difficult to solve. In contrast, the remaining methods involve solving a relatively easier binary classification setting (Class 0 vs. Class 1, Class 2 vs. Class 3, etc.). This could help explain why CLOPS and fine-tuning perform better than MTL. As for GEM, this algorithm involves solving a quadratic programming problem, which simply did not perform well in our healthcare time-series setting.
> > >
> > > We are happy to run additional experiments that may bolster your confidence in the way we have conducted the Class-IL scenario experiments. **Do you have any suggestions on that front?** We had also made the code publicly-available and so you are welcome to experiment with the MTL method, if you so wish.
> > >
> > > We hope the above responses have addressed your concerns.

---

> > > > ### Comment · AnonReviewer5 · 2020-11-19
> > > > **Re: Response to Reviewer 5 - Round 2**
> > > >
> > > > **Contributions**
> > > >
> > > > Please note that the practicality of the problem has never been my concern from the beginning. Nonetheless, I do like the argument about the privacy issue.
> > > >
> > > > My point is, I could not get useful insight from the method or experiments. I think a good paper should provide more than just a method and its scores. If the authors designed a method for physiological data, then the authors should justify their design philosophy, i.e., why they chose a specific architecture for that type of data. Currently, there are an algorithm and scores, but not why.
> > > >
> > > > ---
> > > >
> > > > **Empirical Evidence**
> > > >
> > > > It is hard to understand the authors' response:
> > > > > As a result, in the MTL setting, this problem is a 12-way multi-class classification problem, which is arguably quite difficult to solve. In contrast, the remaining methods involve solving a relatively easier binary classification setting (Class 0 vs. Class 1, Class 2 vs. Class 3, etc.).
> > > >
> > > > In section 3 of the paper, the authors stated:
> > > > > In all of the following cases, task identities are absent during both training and testing and neural architectures are single-headed.
> > > >
> > > > These two clearly contradict each other. If we call a model single-headed, it should classify all 12 classes, not just two for each task. If the architecture is multi-headed, and the task IDs are provided at test time, the authors should test MTL with the same binary classification setting. However, I think single-headed experiments are much more meaningful.
> > > >
> > > > Moreover, the authors did not explain why fine-tuning works better than GEM and MIR, which also suggests insufficient hyperparameter tuning.

---

> > > > > ### Comment · AnonReviewer4 · 2020-11-20
> > > > > **Do public code/data help and could further experiments rescue the manuscript?**
> > > > >
> > > > > The evaluation seems to be the core issue overall, so, as a reviewer with less experience in the continual learning field, I wanted to add my questions here as well, actually more for R5:
> > > > >
> > > > > 1. I agree it is not perfect to have a new algorithm evaluated only on new datasets, but do you consider since the manuscript uses public datasets and public code it might still be valuable enough to publish it so others can run further comparisons?
> > > > > 2. Do you think evaluating the proposed algorithm on common benchmarks would be enough to endorse the manuscript? Or would it depend on the result as well?

---

> > > > > > ### Comment · AnonReviewer5 · 2020-11-20
> > > > > > **Answers to R4**
> > > > > >
> > > > > > Although the evaluation is a critical issue, I put more weight on the weak contribution of this paper. Here are my answers to R4's questions.
> > > > > >
> > > > > > 1. For me, providing code is not a critical factor for the decision. It definitely helps to verify the results, but in this case, the result itself is very confusing.
> > > > > > 2. In my opinion, evaluating on common benchmarks is not enough. If the proposed algorithm performs well on standard benchmarks (although I highly doubt it), then there is no reason to focus only on physiological data, and it should better be introduced as a general CL algorithm. Since the main novelty of this paper is the application of CL to a new domain, I did not ask the authors to put additional effort into testing the standard benchmarks.

---

### Official Review · AnonReviewer1 · 2020-11-06
**interesting problem; unclear task description;**

**Rating:** 4
**Confidence:** 3

**Review:**

The paper presents, CLOPS, a new replay-based continual learning (CL) strategy for physiological signals.

The paper has the following merits. It is well-organized. I can understand its method, different settings of continual learning, and more ablation study on the model parameters. It did through experiments on the 3 datasets and get good results showing CLOPS beats naive baselines like multi-tasks learning, Fine-tuning as well as other CL methods like GEM, MIR.

On the other side, I have the following concerns about the submission.

1. what is the task exactly?

Is cardiac arrhythmia label also time series? Does it have the same sampling frequency as physiological data?
In the Class-IL setting, are you doing multi-class classification or binary class? The notation is confusing. Is task [0,1], task [2,3] meaning binary classifications of class 0 v.s  class 1 and class 2 v.s. class 3? Then what does task [0-2] mean? It appears in the second paragraph of section 6.1.

2. not enough highlight for important parameters (storage and acquisition fractions, b, and a).

Storage and acquisition fractions, "b" and "a" are important parameters of replay-based CL algorithms. However currently, the definition of them is hidden in section 4 and figure captions. They are important parameters. The author even did an ablation study in section 6.4 on it. However, when I first read the paper, after reading the method section, I had no impression of "a" and "b".

3. not well explained experimental result.

In Table 1, it shows Fine-tuning works better than Multi-task learning. It is a bit counter-intuitive to me since MTL has stronger supervision at all the time. Do there exist differences between task difficulty? It seems the task order influences the performance of fine-tuning according to the result in Table 1 and Table 9 in appendix G. Can the author provide any explanation?
In section 6.2, the author argues that CLOPS achieves an AUC after one epoch while fine-tuning requires 20 epochs to achieve that. First, I am not sure whether it is true since it seems the performance of the fine-tuning is not converged yet, its performance may still increasing. Second, clearly, CLOPS has a much larger performance variance than fine-tuning.  So I think the current result does not support that CLOPS is better than fine-tuning.

4. unclear usage of task similarity.

In the appendix, the author shows the task similarity produced by the model. However, I am not sure what we can do with such task similarity. In the abstract, the author claim "this quantification yields insights into both network interpretability and clinical applications". However, in the paper, I can not see any arguments about what insight of network interpretability and clinical applications we can get.

---

> ### Author Response · Authors · 2020-11-18
> **Response to Reviewer 1 - Round 1**
>
> We thank the reviewer for taking the time and effort to review the manuscript and for providing us with valuable feedback. We address your comments below.
>
> **TASK DESCRIPTION**
> In our manuscript, the physiological data are predominantly electrocardiogram (ECG) signals. These are time-series signals that are associated with cardiac arrhythmia labels. As outlined in Appendix B, and as is typical in time-series analysis, we split time-series signals into frames of shorter duration. In our case, each frame was 2500 samples in length which roughly approximates to 5 seconds worth of ECG data (depending on the sampling rate of the dataset). Each of these frames was then associated with a cardiac arrhythmia label.
>
> The Class-IL scenario consists of binary cardiac arrhythmia classification. For example, in Fig. 2 (Sec. 6.1), the first task, [0-1], refers to a binary classification of the two cardiac arrhythmia classes labelled 0 and 1. The second task, [2-3], refers to a binary classification of the two cardiac arrhythmia classes labelled 2 and 3, and so on. Our reference to task [0-2] in the second paragraph of Sec. 6.1 is a typo and should instead state ‘task [2-3]’. To more explicitly address the potential confusion surrounding the specific task at hand (cardiac arrhythmia classification) in addition to the continual learning scenarios, we have included a figure (Fig. 1) to summarize the three continual learning scenarios that we implement.
>
> **STORAGE AND ACQUISITION FRACTIONS**
> We discuss the storage and acquisition fractions in more detail in the methods section to reflect their importance on replay-based continual learning methods. More specifically, we talk about the implications of the storage fraction in Sec. 4.1.2 (last paragraph) and the acquisition fraction in Sec. 4.2.
>
> **FINE-TUNING VS MULTI-TASK LEARNING**
> Typically, one does expect multi-task learning (MTL) to perform better than its continual learning counterpart (see Table 2 Sec 6.3 and Table 8 Appendix E.1). As you mentioned, this is primarily due to the presence of a strong supervisory signal during training. Occasionally, however, a continual learning method and even a naïve fine-tuning one can outperform the MTL setup. This implies that the final set of parameters obtained by the continual learning methods (i.e., at epoch 120 in the Class-IL scenario) achieve stronger generalization performance than the MTL set of parameters. We hypothesize that this improvement in generalization performance can be attributed to positive weight transfer. Such positive weight transfer can be due to the existence of a curriculum (as with curriculum learning) where network generalization improves as a result of presenting tasks of different levels of difficulty to a network in a sequential manner. In fact, we present a set of curriculum learning experiments in Sec. 6.5 to further explore this notion.
>
> As for the effect of task order on the performance of continual learning methods, we hypothesize that one potential explanation lies in the relative difficulty of tasks that are presented sequentially to a network. To support our statement, we direct your attention to Table. 3 (Sec. 6.5) where we order the tasks that are presented to a network based either on a curriculum (easy-to-hard) or an anti-curriculum (hard-to-easy) and observe a discrepancy in the Average AUC (0.744 vs. 0.783).
>
> **EXPLANATION OF TIME-IL RESULTS**
> As for Sec. 6.2 (Time-IL scenario), we claim that CLOPS is better than fine-tuning as it pertains to training efficiency. To substantiate this claim, we direct the reviewer to Fig. 4 (Sec 6.2). At epoch 40, the Fine-tuning method and CLOPS achieve an AUC of 0.50 and 0.62, respectively. Even if we were to consider the lower end of the CLOPS performance, due to the relatively higher variance, CLOPS still achieves an AUC of 0.60, which is higher than its Fine-tuning counterpart. This discrepancy indicates that CLOPS has exhibited positive forward weight transfer. To make an assessment of the training efficiency of CLOPS, we ask the question, “how many epochs of training does it take for the Fine-tuning method to achieve the same AUC (i.e., 0.62) as CLOPS had achieved in a single training epoch?” We find that it takes the Fine-tuning method 20 training epochs (from epoch 40 to epoch 60) to arrive at that AUC value.
>
> **HOW TO USE TASK SIMILARITY**
> As for the purpose of quantifying task-similarity, we direct the reviewer to Sec. 6.5 (paragraph 3). In a nutshell, we exploit task-similarity to order the tasks that are presented to a network and, in turn, help design the curriculum learning procedure. To address this potential confusion more explicitly, we have included in Sec. 6.5 (a) our definition of task-similarity and (b) a figure (Fig. 7) illustrating how we exploited task-similarity to design the task order.
>
> We hope the above responses and the modified version of the manuscript have addressed your concerns.

---

> > ### Comment · AnonReviewer1 · 2020-11-22
> > **New Section 4.2 is nice while other issue remains.**
> >
> > Let me further explain my concerns at each point as the following.
> >
> > TASK DESCRIPTION
> >
> > I am still confused about the Class-IL scenario. From Fig.1 in the revised manuscript, it seems you are using one predictor “f_w” for both binary classification tasks [0-1] and [2-3]. It does not make sense to me. Why we would like to adopt a classifier previously trained to classify cat versus dog to a new task about a car versus plane?
> >
> > STORAGE AND ACQUISITION FRACTIONS
> >
> > The new section 4 looks better.
> >
> > FINE-TUNING VS MULTI-TASK LEARNING
> >
> > The author is arguing that “the final set of parameters obtained by the continual learning methods (i.e., at epoch 120 in the Class-IL scenario) achieve stronger generalization performance than the MTL set of parameters. ” Current result in Tab.1 is not enough to support the argument. We need the training error of “Fine-tuning” and “MTL” to compare the generalization. I would like to know whether “MTL” has a similar training error as the “Fine-tuning”. If “MTL” has a worse training error than “Fine-tuning”, it is likely that the “MTL” is underfitting due to potential reasons like tasks have interference. It is a standard issue of “MTL”. One could consider many tricks like PCGrad [1] to mitigate it.
> >
> > By the way, it related to R5’s concern “I think this strongly suggests that the baselines are poorly tuned”. At a high-level, I agree with R5. It seems the author still needs some effort here.
> >
> > [1] Yu, Tianhe, et al. "Gradient surgery for multi-task learning." arXiv preprint arXiv:2001.06782 (2020).
> >
> > EXPLANATION OF TIME-IL RESULTS
> >
> > I understand that from Fig.4, it looks that “Fine-tuning” takes 20 epochs to reach the performance of “CLOPS” at epoch 1. However, my main concern is that learning is not finished yet. The curve of “Fine-tuning” has a much large slope comparing to “CLOPS”. The trend continuing persists for like 100 epochs, the “Fine-tuning” will be a clear winner against “CLOPS”.  So I suggest the author train the model longer and draw conclusions after the model converged.
> >
> > HOW TO USE TASK SIMILARITY
> >
> > From the current result presented in Tab 3, I cannot agree that curriculum learning or anti curriculum learning is performing better than “Random”.  Clearly, “Random” has a higher AUC. So I am not sure about the benefit of doing such task-similarity-based curriculum learning.
> >
> > In summary, since most of my concerns are still there, my score remains unchanged.

---

### Author Response · Authors · 2020-11-23
**Final Version of Manuscript**

We thank all the reviewers for taking the time and effort to read the manuscript and for providing us with valuable feedback.

**FINAL UPLOAD OF MANUSCRIPT**
We have now modified the manuscript, as per your comments, and uploaded the main manuscript and the supplementary material. Here are the high-level changes that we have made to the manuscript:

1) We have included Fig. 1 in **Sec. 3.1** to clarify the various continual learning scenarios that we are experimenting under.
2) We have modified **Sec. 4** significantly to improve clarity as it pertains to the explanation of the loss-weighting mechanism and the motivation behind the storage function (Eq. 6).
3) We have explicitly mentioned the storage and acquisition fractions in **Sec. 4**.
4) We have modified the figures containing learning curves (e.g., Fig. 3) to improve clarity and illustrate when the network is in a 'training' phase. This is indicated by the coloured-blocks.
5) We have modified **Sec. 6.2** to extend the training on task Term 3 until convergence. We find that CLOPS not only exhibits forward weight transfer (FWT) but also mitigates catastrophic forgetting relative to the fine-tuning setup. More details can be found in Sec. 6.2.
6) We conducted Random Storage and Acquisition experiments in **Sec. 6.4** in order to better illustrate the useful components of CLOPS. These results can be seen in Fig. 5.
7) We have annotated the ECG recordings in Fig. 6 in order to better illustrate our point regarding ST-elevation (a medical condition).
8) We have modified **Sec. 6.5** to better illustrate how we go about quantifying task similarity and leveraging that to generate a curriculum.

We hope our responses to your reviews in addition to the modified version of the manuscript have addressed your concerns.

---

### Decision · Program_Chairs · 2021-01-07
**Final Decision**

**Decision:**

Reject

**Comment:**

Reviewers could not reach consensus here and important concerns from one reviewer on empirical results could not be convincingly addressed. The authors have provided a comprehensive response to the reviews, yet failed to convince them.